# MIDAS: Misalignment-based Data Augmentation Strategy for Imbalanced Multimodal Learning

**Seong-Hyeon Hwang**[*]  **Soyoung Choi**[*]  **Steven Euijong Whang**[†]
KAIST
{sh.hwang, wer07081, swhang}@kaist.ac.kr

## Abstract

Multimodal models often over-rely on dominant modalities, failing to achieve optimal performance. While prior work focuses on modifying training objectives or optimization procedures, data-centric solutions remain underexplored. We propose MIDAS, a novel data augmentation strategy that generates misaligned samples with semantically inconsistent cross-modal information, labeled using unimodal confidence scores to compel learning from contradictory signals. However, this confidence-based labeling can still favor the more confident modality. To address this within our misaligned samples, we introduce *weak-modality weighting*, which dynamically increases the loss weight of the least confident modality, thereby helping the model fully utilize weaker modality. Furthermore, when misaligned features exhibit greater similarity to the aligned features, these misaligned samples pose a greater challenge, thereby enabling the model to better distinguish between classes. To leverage this, we propose *hard-sample weighting*, which prioritizes such semantically ambiguous misaligned samples. Experiments on multiple multimodal classification benchmarks demonstrate that MIDAS significantly outperforms related baselines in addressing modality imbalance.

## 1 Introduction

The human ability to perceive and interpret the world is inherently multimodal, integrating information from visual, auditory, and textual cues to form a complete understanding. Motivated by this human perception, multimodal learning has gained significant attention in artificial intelligence research [1, 2]. This approach has led to breakthroughs across domains, including vision-language modeling [3], medical diagnosis [4], and autonomous systems [5]. In spite of its success in various areas, one of the fundamental challenges is imbalanced multimodal learning [6], where models tend to rely on the more informative modality while neglecting the weaker ones. Modality imbalance leads models toward degraded overall performance, even worse than unimodal models [7].

While extensive recent studies [7, 8, 9, 10] have addressed the multimodal imbalance problem, they have largely overlooked the importance of data and feature processing. Most studies solve the problem through designing new training objectives or optimization strategies using only aligned samples, such as re-weighting each modality importance [7] or adjusting the direction and magnitude of gradients [11]. Even if a few studies propose data or feature-level solutions, these approaches typically rely on masking [12] or zero-vector substitution [13]. Such approaches intentionally limit information use and fail to fully exploit the information embedded in multimodal data.

To overcome this limitation, we introduce a different data-centric perspective: leveraging misaligned samples as informative data that can reveal and help address modality imbalance, rather than as noise or outliers [14]. A misaligned sample is formed by combining modalities from different original

---

[*]Equal contribution.   [†]Corresponding author.

39th Conference on Neural Information Processing Systems (NeurIPS 2025).

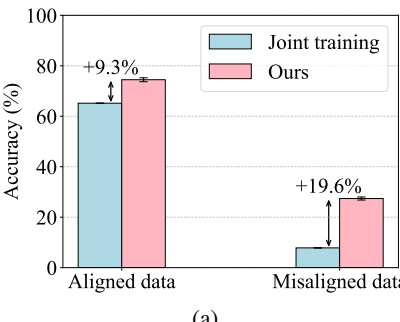
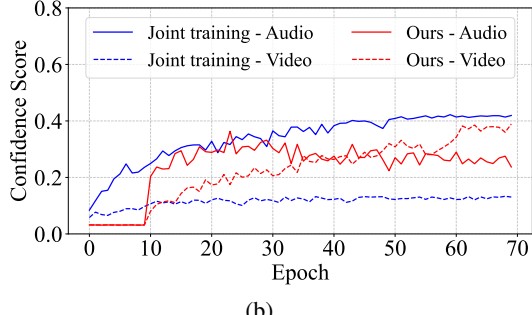

|                          |                          |
|--------------------------|--------------------------|
| (a)                      | (b)                      |

Figure 1: (a) Accuracy comparison between Joint training and our method on aligned (original) and misaligned validation data. (b) Comparison of modality confidence scores between Joint training and our method when predicting misaligned validation data on the Kinetics-Sounds dataset.

samples (e.g., an image of a dog paired with text describing a cat), with its original labels. This structure preserves the features of the original samples, enabling full exploitation of multimodal representations. More importantly, because aligned samples are associated with a single label shared across all modalities, it is difficult to determine whether the model is truly using all modalities or relying on just one. In contrast, misaligned samples contain modality-specific information pointing to different labels, making it possible to diagnose modality imbalance by analyzing the model's prediction. Ideally, a model trained to truly leverage information across modalities should be able to identify content related to all modalities, even when the input is synthetically misaligned.

To investigate whether a standard multimodal model has this capability, we evaluate its accuracy on aligned and misaligned samples from the Kinetics-Sounds dataset [15]. For aligned samples, accuracy is measured using the top-1 predicted class, while for misaligned samples, it is measured based on whether both ground-truth labels appear within the top-2 predictions. As shown in Figure 1a, the multimodal model exhibits low accuracy on misaligned samples (6.3%) compared to aligned data (65.5%), revealing its biased reliance on a specific modality. Furthermore, Figure 1b shows that the model consistently predicts the class associated with the dominant modality (audio, in this case) with high confidence when facing misaligned inputs.

Thus, we propose MIDAS, a novel modality-agnostic multimodal data augmentation strategy designed to mitigate modality imbalance (see Figure 2). MIDAS systematically generates misaligned samples by pairing modalities from different source instances, where each modality has different labels. The key idea is to enforce the model to recognize conflicting signals within a single input. To assign a meaningful label for misaligned samples, we take a weighted average of labels based on the confidence scores of each unimodal classifier, since the modalities are not equally informative [10].

However, the model still relies heavily on the stronger modality, as more confident modalities contribute more to the target label. To supplement this, we first propose *weak-modality weighting*, which increases the loss contribution of the least-confident modality. This adjustment counteracts the weakness of unimodal confidence-based labeling and allows the model to better attend to underutilized sources. Moreover, not all misaligned samples are equally useful. Higher similarity between the swapped and original embeddings makes the semantic conflict more subtle, causing these samples to be more challenging and informative. Thus, we employ *hard-sample weighting*, which emphasizes such difficult samples to improve class discrimination. Together, these techniques enhance the model's ability to learn balanced representations from misaligned data.

We conduct comprehensive evaluations of MIDAS on multiple real-world multimodal datasets for classification. The results demonstrate that MIDAS effectively enhances modality utilization and outperforms existing approaches for addressing imbalanced multimodal learning. Our findings highlight that MIDAS is the first data augmentation method to construct misaligned pairs with distinct labels and advanced weighting techniques.

**Summary of contributions:**   (1) We propose a new modality-agnostic data augmentation method based on misalignment for multimodal imbalance learning; (2) We introduce a novel labeling strategy and two weighting mechanisms to maximize learning from misaligned data; (3) We show that MIDAS outperforms related baselines via extensive experiments.

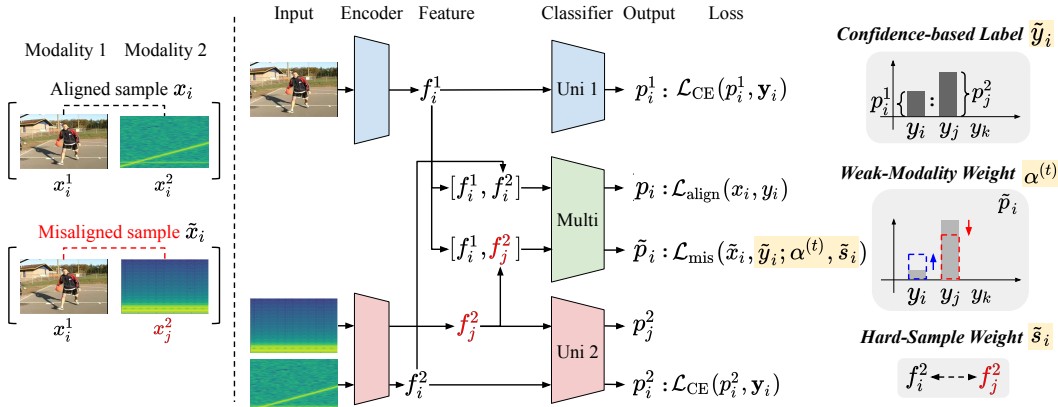

Figure 2: MIDAS trains a multimodal model on both aligned and misaligned samples with conflicting semantics simultaneously. MIDAS consists of three main components: 1) We label misaligned samples with a confidence-based labeling strategy using unimodal classifiers. 2) Weak-modality weighting increases the loss weight of the least confident modality. 3) Hard-sample weighting assigns a higher loss weight to more confusing misaligned samples containing similar semantics.

## 2 Related work

**Imbalanced Multimodal Learning** The disparity in learning effectiveness between modalities is a key challenge in multimodal learning. Many studies [7, 16, 17, 18, 6, 11, 19, 20, 21, 22] have highlighted this problem with two representative directions using only aligned samples: optimization-based strategies that adjust training dynamics, and data/feature-based approaches that manipulate inputs or representations. Optimization-based techniques such as AGM [8] and OGM [16] aim to reduce the gradient strength for dominant modalities based on performance or Shapley value estimations. However, they often introduce additional overhead through complex steps, including repeating gradient computation. In parallel, data and feature-centric approaches like selective resampling [13] and adaptive masking [12] aim to support weak modality by generating weak modality-specific data or masking dominant modality features, respectively. While effective, they often perturb data or features and fail to fully leverage information from the original data. Recently, a few studies have used misaligned pairs to inject cross-modal information. LFM [21] leverages misaligned data points as negative samples in contrastive learning, and MCR [23] employs within-batch permutations to estimate conditional mutual information. However, these methods remain unsupervised learning schemes, limiting their applicability to downstream tasks. In contrast, our approach treats misaligned data as supervised training signals, enabling us to leverage the original features in addition to the misaligned relationship between modalities for imbalanced multimodal learning.

**Multimodal Data Augmentation** Although data augmentation has greatly improved unimodal learning, its application to multimodal settings remains relatively underdeveloped. Recently, a few studies [24, 25, 26, 27] have focused on augmentation techniques for multimodal data. Yet many of these methods are either modality-specific [25, 28, 29] or apply uniform transformation to all modalities [27], failing to account for the varying influences that different modalities exert on learning. For instance, MixGen [25] only augments image-text pairs by interpolating images and concatenating the corresponding texts. In contrast, PowMix [24] introduces a regularization strategy that interpolates latent representations with anisotropic mixing and adjusts their contributions. However, it does not explicitly address modality imbalance. On the other hand, our method focuses on imbalanced multimodal learning, while considering diverse and complementary cross-modal information [1].

## 3 Method

In this section, we detail our approach for solving the multimodal imbalance problem. We first introduce the basic setup and notation in Sec. 3.1. Then, we describe our core mechanisms generating misaligned samples (Sec. 3.2), unimodal confidence based sample-level labeling (Sec. 3.3), weak-modality weighting (Sec. 3.4), and hard-sample weighting (Sec. 3.5). Finally, we present the overall

training objective combining these elements in Sec. 3.6. For simplicity, we consider a two-modality setting (M=2) in this paper, such as image-text pairs. We briefly discuss how these concepts generalize to scenarios with more than two modalities in the Appendix (Sec. A.2).

## 3.1 Preliminaries

We consider a dataset $D = \{(x_i, y_i)\}_{i=1}^N$, where each input $x_i = (x_i^1, \ldots, x_i^M)$ consists of $M$ modalities and $y_i \in \{1, \ldots, C\}$ is the class label. Each modality $x_i^m$ is mapped to a feature $f_i^m = \varphi_m(x_i^m)$ by a dedicated encoder $\varphi_m$; the set of encoders is $\varphi = \{\varphi_m\}_{m=1}^M$. These features $(f_i^1, \ldots, f_i^M)$ are processed by a multimodal classification layer $g_f$ to produce logits $z_i = g_f(f_i^1, \ldots, f_i^M)$, yielding predicted probabilities $p_i = \mathrm{softmax}(z_i)$. In parallel, $M$ unimodal classification layers $\{g_m\}_{m=1}^M$ process individual features $f_i^m$ to yield unimodal logits $z_i^m = g_m(f_i^m)$ and probabilities $p_i^m = \mathrm{softmax}(z_i^m)$. Crucially, the encoders $\varphi$ are shared across both multimodal and unimodal pathways. Unless stated otherwise, we use the standard cross-entropy loss, $\mathcal{L}_{\mathrm{CE}}$, for training.

## 3.2 Generating Misaligned Samples

To reduce the tendency of the model to over-rely on dominant modality, we systemically generate *misaligned samples* that contain conflicting semantic information across modalities. Multimodal models trained only on aligned data with hard labels often learn "shortcuts", focusing excessively on the most informative modality while ignoring the others [7, 21]. To address this imbalance, we explicitly leverage such misaligned samples, which have generally been treated as noise or outliers [14]. By requiring the model to interpret information from all modalities within these misaligned samples, we encourage the model to develop a more balanced reliance on each modality.

Consider an aligned sample $x_i = (x_i^1, x_i^2)$ with label $y_i$. Within the same mini-batch, we randomly select another sample $x_j = (x_j^1, x_j^2)$ such that $y_j \neq y_i$. We adopt random replacement due to its two advantages: computational efficiency and improved model generalizability. To substantiate this choice, we provide comparisons with alternative misaligned-sample generation strategies in Appendix (Sec. A.4). A misaligned sample $\tilde{x}_i$ is then constructed by swapping one modality as follows:

$$\tilde{x}_i = (\tilde{x}_i^1, \tilde{x}_i^2) = (x_i^1, x_j^2) \tag{1}$$

This $\tilde{x}_i$ combines the first modality from sample $i$ with the second modality from sample $j$. For example, if $x_i^1$ is an image of a "cat" ($y_i$ = "cat") and $x_j^2$ is a text description of a "dog" ($y_j$ = "dog"), the misaligned sample $\tilde{x}_i = (x_i^1, x_j^2)$ pairs the "cat" image with the "dog" text. Symmetrically, we also generate and utilize $(x_j^1, x_i^2)$.

## 3.3 Unimodal Confidence based Sample-level Labeling

Supervising the model with the generated misaligned sample $\tilde{x}_i$ presents a challenge: determining an appropriate target label for $\tilde{x}_i$. Thus, we generate a label $\tilde{y}_i \in \mathbb{R}^C$ for each $\tilde{x}_i$ by using a sample-level labeling strategy. Since $\tilde{x}_i$ combines two modalities from different source classes, using a single hard label for either class is inappropriate. On the other hand, a naïve average of source labels ignores valuable supervision or inadvertently overweights a less informative modality [10].

Our key idea is to compute $\tilde{y}_i$ by evaluating how confidently each unimodal classifier predicts the original label associated with its respective modality. Specifically, we use the unimodal classifiers ($g_1$ and $g_2$) to estimate the confidence scores for their corresponding source labels, based on the individual components ($x_i^1$ and $x_j^2$) of the misaligned sample. This approach draws inspiration from prior work that utilizes unimodal output probabilities to estimate modality importance or reliability [11, 30]. Furthermore, sample-level labeling is crucial because even within the same modality, the amount of discriminative information can vary across samples [13].

Given a misaligned sample $\tilde{x}_i = (x_i^1, x_j^2)$, we obtain the unimodal output probabilities $p_i^1 = \mathrm{softmax}(g_1(\varphi_1(x_i^1)))$ and $p_j^2 = \mathrm{softmax}(g_2(\varphi_2(x_j^2)))$. Using the probabilities, we calculate the confidence scores for the original class labels of each modality: $(p_i^1)_{y_i}$ (confidence of modality 1 on label $y_i$) and $(p_j^2)_{y_j}$ (confidence of modality 2 on label $y_j$), where $(\cdot)_k$ indicates $k$-th components of the input. These confidences are normalized as:

$$\tilde{c}_i^1 = \frac{(p_i^1)_{y_i}}{(p_i^1)_{y_i} + (p_j^2)_{y_j}}, \quad \tilde{c}_i^2 = \frac{(p_j^2)_{y_j}}{(p_i^1)_{y_i} + (p_j^2)_{y_j}} \tag{2}$$

Then, the target label $\tilde{y}_i$ is a weighted average of the one-hot encoded source labels as follows:

$$\tilde{y}_i = \sum_{m=1}^{M=2} \tilde{c}_i^m \mathbf{y}_{i,m} = \tilde{c}_i^1 \mathbf{y}_i + \tilde{c}_i^2 \mathbf{y}_j \tag{3}$$

where $\mathbf{y}_i, \mathbf{y}_j \in \{0,1\}^C$ are the one-hot vectors for labels $y_i$ and $y_j$, respectively. For example, if $(p_i^1)_{y_i} = 0.9$ and $(p_j^2)_{y_j} = 0.3$, then the normalized confidences are $\tilde{c}_i^1 = 0.75$ and $\tilde{c}_i^2 = 0.25$. In this case, the target label is $\tilde{y}_i = 0.75 \mathbf{y}_i + 0.25 \mathbf{y}_j$. The $\tilde{y}_i$ reflects the relative contribution of each modality in the misaligned sample, as estimated by the unimodal classifiers. Then, the loss for misaligned sample $\tilde{x}_i$ with its label $\tilde{y}_i$ is defined as:

$$\mathcal{L}_{\text{mis}}(\tilde{x}_i, \tilde{y}_i) = \mathcal{L}_{\text{CE}}(\tilde{p}_i, \tilde{y}_i) = -\sum_{c=1}^{C} \left( \tilde{c}_i^1 (\mathbf{y}_i)_c + \tilde{c}_i^2 (\mathbf{y}_j)_c \right) \log((\tilde{p}_i)_c) \tag{4}$$

where $\tilde{p}_i = \text{softmax}\left(g_f(\varphi(\tilde{x}_i))\right)$ is the output probability of the multimodal classifier for $\tilde{x}_i$. In practice, we implement a warm-up phase to train encoders and unimodal classifiers for each modality before the labeling process. During this phase, encoders and unimodal classifiers are trained with aligned data to ensure that confidence scores reflect meaningful modality reliability. Without this step, unreliable confidence estimates could mislead the labeling and downstream weighting process during the early stages of training.

### 3.4 Weak-Modality Weighting

To further prevent the underrepresented modality from being overshadowed in misaligned samples, we introduce a weak-modality weight $\alpha = \{\alpha_1, \alpha_2\}$. While the target label $\tilde{y}_i$ provides a supervisory signal for the misaligned sample $\tilde{x}_i$, a standard cross-entropy loss $\mathcal{L}_{\text{CE}}(\tilde{p}_i, \tilde{y}_i)$ may still be insufficient to suppress the model's tendency to rely on the dominant modality. Even when optimizing toward $\tilde{y}_i = \tilde{c}_i^1 \mathbf{y}_i + \tilde{c}_i^2 \mathbf{y}_j$, the multimodal model $g_f$ could primarily focus on the component of the modality with higher confidence (i.e., $\max(\tilde{c}_i^1, \tilde{c}_i^2)$), potentially undervaluing the signal of the other modality. Thus, the weak-modality weight dynamically increases the loss weight for the least confident modality when the multimodal model's normalized confidence for the class associated with that modality falls below the corresponding unimodal confidence used to generate the target label.

We first identify the *least confident modality* $\hat{m}$ whose unimodal classifier shows the lowest average confidence across a batch of misaligned samples $\tilde{B}$ as follows:

$$\hat{m} = \underset{m \in \{1,2\}}{\arg\min} \, \mathbb{E}_{(\tilde{x}_i, \tilde{y}_i) \sim \tilde{B}}[\tilde{c}_i^m] \tag{5}$$

We initialize weights $\alpha_1^{(t)}, \alpha_2^{(t)}$ to 1, and only update the weight of the identified least confident modality, $\alpha_{\hat{m}}^{(t)}$ at each batch iteration $t$. This update mechanism compares the target label associated with modality $\hat{m}$ (i.e., $\tilde{c}_i^{\hat{m}}$) within the misaligned label against the multimodal model's predicted confidence for the corresponding source class from $\tilde{x}_i$. Let $\tilde{p}_i = \text{softmax}(g_f(\varphi(\tilde{x}_i)))$ be the multimodal prediction for $\tilde{x}_i$, and $\tilde{y}_i^{(\hat{m})}$ be the target label for the modality $\hat{m}$ in the sample $\tilde{x}_i$ (e.g., if $\tilde{x}_i = (x_i^1, x_j^2)$, then $\tilde{y}_i^{(1)} = y_i$ and $\tilde{y}_i^{(2)} = y_j$). To accurately assess the model's confidence for the specific source class associated with the modality $\hat{m}$ in the misaligned sample, we calculate the normalized multimodal confidence for class $\tilde{y}_i^{\hat{m}}$ as $(\tilde{c}_i)_{\tilde{y}_i^{\hat{m}}} = (\tilde{p}_i)_{\tilde{y}_i^{\hat{m}}} / ((\tilde{p}_i)_{y_i} + (\tilde{p}_i)_{y_j})$. The update signal $\Delta_\alpha$ is computed as the batch-averaged difference between the target label and this normalized predicted confidence:

$$\Delta_\alpha = \text{sign}\left( \mathbb{E}_{\tilde{x}_i \sim \tilde{B}}[\tilde{c}_i^{\hat{m}}] - \mathbb{E}_{\tilde{x}_i \sim \tilde{B}}[(\tilde{c}_i)_{\tilde{y}_i^{\hat{m}}}] \right) \tag{6}$$

where the expectations are over the current batch of misaligned samples $\tilde{B}$. The weight $\alpha_{\hat{m}}^{(t+1)}$ is updated as follows, while the weight for the other modality remains 1:

$$\alpha_{\hat{m}}^{(t+1)} = \max\left(1, \alpha_{\hat{m}}^{(t)} + \eta \cdot \Delta_\alpha\right); \quad \alpha_{m \neq \hat{m}}^{(t+1)} = 1 \tag{7}$$

where $\eta > 0$ is the step size. If the model under-predicts the contribution of modality $\hat{m}$ (i.e., $\Delta_\alpha > 0$), $\alpha_{\hat{m}}$ increases, thereby amplifying its influence in the misaligned sample loss. For example, if the normalized unimodal confidence of the least confident modality $\hat{m}$ is $\tilde{c}_i^{\hat{m}} = 0.25$, and the normalized multimodal confidence for its corresponding class is only $(\tilde{c}_i)_{\tilde{y}_i^{\hat{m}}} = 0.10$, the multimodal model underestimates the importance of the modality $\hat{m}$. In this case, $\alpha_{\hat{m}}$ becomes larger than 1. The loss for misaligned samples with $\alpha^{(t)}$ will be presented in Sec. 3.6.

## 3.5 Hard-Sample Weighting

We also propose a hard-sample weight $\tilde{s}_i$, which modulates the influence of a misaligned sample $\tilde{x}_i$. The intuition is that not all misaligned samples are equally informative. If the swapped-in feature is highly similar to the original feature despite originating from a different class, the semantic conflict might be difficult for the model to discern. Thus, we encourage the model to focus more on the misaligned samples composed of more similar semantics, thereby improving its capacity to capture fine-grained feature representations. The hard-sample weight is based on how similar the feature embedding of the swapped modality is to that of its original counterpart from the same source sample.

Consider the misaligned sample $\tilde{x}_i = (x_i^1, x_j^2)$, where $x_j^2$ (from sample $j$, label $y_j$) replaces the original $x_i^2$ (from sample $i$, label $y_i \neq y_j$). We compare the feature vector of the original modality, $f_i^2 = \varphi_2(x_i^2)$, with the feature vector of the swapped-in modality, $f_j^2 = \varphi_2(x_j^2)$. The $\tilde{s}_i$ is calculated as the cosine similarity between these two features:

$$\tilde{s}_i = \frac{(f_i^2)^\top f_j^2}{\|f_i^2\|_2 \|f_j^2\|_2} \tag{8}$$

This weight $\tilde{s}_i \in [-1, 1]$ quantifies how similar the swapped-in modality feature ($f_j^2$) is to the feature it replaced ($f_i^2$). A higher $\tilde{s}_i$ suggests a more subtle semantic difference between the components involved in the swap for the second modality. As detailed later in Sec. 3.6, this weight modulates the loss contribution of the misaligned samples. (If the other type of misaligned sample $\tilde{x}_i = (x_j^1, x_i^2)$ is generated, the similarity would be calculated between $f_i^1$ and $f_j^1 = \varphi_1(x_j^1)$). The harder the misaligned sample, the higher its loss weight.

## 3.6 Overall Training Objective

Our final training objective combines the standard supervised loss from aligned samples with the weighted supervised loss from misaligned samples. For aligned data $(x_i, y_i)$, the loss $\mathcal{L}_{\text{align}}$ for multimodal model and $\mathcal{L}_{\text{uni}}$ for unimodal models are represented as:

$$\mathcal{L}_{\text{align}}(x_i, y_i) = \mathcal{L}_{\text{CE}}(p_i, \mathbf{y}_i); \quad \mathcal{L}_{\text{uni}}(x_i, y_i) = \mathcal{L}_{\text{CE}}(p_i^1, \mathbf{y}_i) + \mathcal{L}_{\text{CE}}(p_i^2, \mathbf{y}_i) \tag{9}$$

As mentioned in Sec. 3.3, we train encoders and unimodal classifiers prior to training multimodal classifiers using $\mathcal{L}_{\text{uni}}$ during the warm-up phase.

The final objective $\mathcal{L}_{\text{mis}}$ for a misaligned sample $\tilde{x}_i$ (i.e., $(x_i^1, x_j^2)$) is calculated using the confidence-based label $\tilde{y}_i$, the weak-modality weight $\alpha^{(t)}$, and the hard-sample weight $\tilde{s}_i$ as follows:

$$
\begin{aligned}
\mathcal{L}_{\text{mis}}(\tilde{x}_i, \tilde{y}_i; \alpha^{(t)}, \tilde{s}_i) &= (1 + \frac{\tilde{s}_i + 1}{2}) \cdot \mathcal{L}_{\text{CE}}(\tilde{p}_i, \tilde{y}_i; \alpha^{(t)}) \\
&= (1 + \frac{\tilde{s}_i + 1}{2}) \cdot \left[ -\sum_{c=1}^{C} \left( \alpha_1^{(t)} \tilde{c}_i^1(\mathbf{y}_i)_c + \alpha_2^{(t)} \tilde{c}_i^2(\mathbf{y}_j)_c \right) \log((\tilde{p}_i)_c) \right]
\end{aligned}
\tag{10}
$$

The total loss $\mathcal{L}_{\text{total}}$ is averaged over a mini-batch $B$:

$$\mathcal{L}_{\text{total}} = \frac{1}{|B|} \sum_{(x_i, y_i) \in B} \left[ \mathcal{L}_{\text{align}}(x_i, y_i) + \mathcal{L}_{\text{uni}}(x_i, y_i) + \lambda \mathcal{L}_{\text{mis}}(\tilde{x}_i, \tilde{y}_i; \alpha^{(t)}, \tilde{s}_i) \right] \tag{11}$$

The generation of $\tilde{x}_i$, the computation of $\tilde{y}_i$ and $\tilde{s}_i$, and the update of $\alpha^{(t)}$ occur dynamically within the training loop. The model parameters ($\{\varphi_m\}, g_f, \{g_m\}$) are updated by minimizing $\mathcal{L}_{\text{total}}$. The overall algorithm of our method is given in the Appendix (Sec. A.1).

In addition, we analyze the computational complexity of MIDAS. For instance, in a two-modality setting ($M = 2$), MIDAS generates two misaligned samples per original sample $x_i$. As this adds only a constant number of additional samples per $x_i$, the overall computational complexity remains $O(N)$, where $N$ is the number of training samples. While increasing the number of modalities $M$ would lead to generating more samples, $M$ is typically small (2 or 3) in real-world settings.

# 4 Experiments

We provide experimental results for MIDAS, evaluating its performance on multimodal classification tasks in the presence of the imbalance modality problem. We report the mean and standard deviation ($\pm$) across three independent runs with different random seeds for all experiments. All experiments are conducted using NVIDIA GeForce RTX A6000 and Quadro RTX 8000 GPUs.

## 4.1 Experimental Settings

**Datasets** We evaluate our method and baselines on four widely used benchmarks for imbalanced multimodal learning, each exhibiting varying degrees and types of modality characteristics: Kinetics-Sounds [15] is a dataset linking audio and video clips for action recognition with 31 classes. CREMA-D [31] is an audiovisual dataset for emotion recognition featuring actors speaking sentences with 6 classes. UCF-101 [32] is an action recognition dataset consisting of RGB frames and optical flows with 101 classes. Food-101 [33] is a dataset of food images paired with their corresponding textual recipes with 101 classes. Additional dataset statistics are summarized in the Appendix (Sec. A.3).

**Metrics** We report the Top-1 Accuracy (Acc) and F1-Score (F1) as our primary evaluation metrics in percentages following [6]. Accuracy measures the overall classification correctness, while F1-Score provides a balanced measure between precision and recall, which is particularly relevant in cases of class imbalance or varying difficulty across classes. For both metrics, higher is better.

**Implementation Details** We conduct experiments following the configurations in [6]. For Kinetics-Sounds and CREMA-D, we use ResNet-18 [34] encoders for both audio and video, training from scratch. For UCF-101, we also use ResNet-18 as encoders. For the Food-101 dataset, we use a pre-trained ResNet-18 and a pre-trained ELECTRA [35] as image and text encoders, respectively. More detailed configurations are provided in the Appendix (Sec. A.3).

**Baselines** We compare MIDAS against the following related baselines, which manipulate features or generate new types of data for imbalanced multimodal learning: 1) *Joint training* is a vanilla multimodal learning technique; 2) *SMV* [13] re-samples data from low-contributing modalities based on sample-level modality valuation; 3) *OPM* [36] modulates features by weight multiplication to adjust the contribution of each modality by predicting per-sample modulation weights during training; 4) *AMCo* [12] applies adaptive masking on the features of the dominant modality to adjust the learning difficulty; 5) *LFM* [21] combines multimodal learning with contrastive learning through dynamic integration; 6) *MCR* [23] leverages misaligned features for unsupervised learning and uses game-theoretical regularization to balance the contributions of modalities.

## 4.2 Comparison with Baselines

We compare the performance of MIDAS with existing related baselines across four benchmark datasets. As shown in Table 1, MIDAS consistently outperforms the baselines across all datasets in both accuracy and F1-score. Specifically, MIDAS achieves significant improvements in accuracy on the Kinetics-Sounds (3.13%p↑) and CREMA-D (4.08%p↑) compared to the best-performing baselines trained solely on aligned samples. Furthermore, MIDAS is effective not only on audio-video pairs (representative examples of imbalanced modality pairs) but also on other modalities, including image, text, and optical flow, demonstrating its versatility. While MIDAS shows comparable performance to SMV on the UCF-101, it is worth noting that SMV has more epochs and opportunity to learn due to approximately 3x more generated data. In addition, compared to LFM and MCR, which utilize misaligned samples in unsupervised or contrastive learning frameworks, MIDAS achieves superior performance through supervised learning. By providing explicit labels derived from unimodal confidences, MIDAS offers direct supervision to the model, enabling more effective representation learning compared to these less direct (unsupervised or contrastive) approaches.

## 4.3 Ablation Study

To understand the contributions of each component in MIDAS, we evaluate the impact of three key elements: the warm-up phase (W) for unimodal classifiers, weak-modality weighting (WM), and hard-sample weighting (HS) on all datasets. As shown in Table 2, each component individually

Table 1: Performance results comparing MIDAS against related baselines on four multimodal datasets. The best and second-best results are highlighted in bold and underlined, respectively.

| Method | Kinetics-Sounds | | CREMA-D | | UCF-101 | | Food-101 | |
|---|---|---|---|---|---|---|---|---|
| | Acc (↑) | F1 (↑) | Acc (↑) | F1 (↑) | Acc (↑) | F1 (↑) | Acc (↑) | F1 (↑) |
| Joint training | $63.92_{\pm0.23}$ | $55.54_{\pm0.30}$ | $60.28_{\pm1.16}$ | $58.60_{\pm1.14}$ | $90.07_{\pm1.23}$ | $83.80_{\pm1.56}$ | $91.35_{\pm0.17}$ | $85.75_{\pm0.35}$ |
| SMV [13] | $65.76_{\pm1.48}$ | $57.59_{\pm1.56}$ | $67.94_{\pm1.73}$ | $66.83_{\pm1.89}$ | $\mathbf{95.24}_{\pm0.39}$ | $\mathbf{91.80}_{\pm0.88}$ | $91.64_{\pm0.28}$ | $86.26_{\pm0.54}$ |
| OPM [36] | $67.35_{\pm0.67}$ | $59.29_{\pm0.70}$ | $63.97_{\pm1.72}$ | $62.71_{\pm1.68}$ | $91.73_{\pm0.51}$ | $86.42_{\pm0.66}$ | $\underline{92.40}_{\pm0.27}$ | $\underline{87.41}_{\pm0.59}$ |
| AMCo [12] | $67.04_{\pm0.68}$ | $58.41_{\pm1.05}$ | $69.91_{\pm3.23}$ | $68.85_{\pm3.28}$ | $93.77_{\pm0.53}$ | $89.46_{\pm0.79}$ | $92.00_{\pm0.23}$ | $86.79_{\pm0.47}$ |
| LFM [21] | $64.88_{\pm0.61}$ | $56.39_{\pm1.00}$ | $64.02_{\pm1.82}$ | $62.50_{\pm2.24}$ | $91.86_{\pm0.41}$ | $86.48_{\pm0.61}$ | $92.15_{\pm0.34}$ | $86.48_{\pm0.49}$ |
| MCR [23] | $\underline{71.75}_{\pm0.56}$ | $\underline{64.23}_{\pm0.69}$ | $\underline{70.91}_{\pm1.22}$ | $\underline{70.19}_{\pm1.17}$ | $91.84_{\pm0.27}$ | $86.27_{\pm0.46}$ | $90.58_{\pm0.26}$ | $84.62_{\pm0.38}$ |
| **MIDAS** | $\mathbf{74.88}_{\pm0.49}$ | $\mathbf{67.18}_{\pm1.23}$ | $\mathbf{74.99}_{\pm0.31}$ | $\mathbf{73.82}_{\pm0.33}$ | $\underline{95.20}_{\pm0.18}$ | $\underline{91.66}_{\pm0.57}$ | $\mathbf{93.46}_{\pm0.36}$ | $\mathbf{89.02}_{\pm0.62}$ |

Table 2: Ablation study on four multimodal datasets.

| W | WM | HS | Kinetics-Sounds | | CREMA-D | | UCF-101 | | Food-101 | |
|---|---|---|---|---|---|---|---|---|---|---|
| | | | Acc (↑) | F1 (↑) | Acc (↑) | F1 (↑) | Acc (↑) | F1 (↑) | Acc (↑) | F1 (↑) |
| ✗ | ✗ | ✗ | $71.70_{\pm0.61}$ | $63.47_{\pm0.36}$ | $72.32_{\pm0.52}$ | $70.94_{\pm3.14}$ | $94.16_{\pm0.08}$ | $90.17_{\pm0.32}$ | $93.39_{\pm0.19}$ | $88.91_{\pm0.48}$ |
| ✓ | ✗ | ✗ | $72.32_{\pm0.52}$ | $64.04_{\pm0.81}$ | $71.25_{\pm1.87}$ | $70.28_{\pm1.79}$ | $94.97_{\pm0.41}$ | $91.40_{\pm0.97}$ | $93.46_{\pm0.32}$ | $89.02_{\pm0.48}$ |
| ✗ | ✓ | ✗ | $72.84_{\pm0.41}$ | $64.76_{\pm0.34}$ | $72.28_{\pm1.14}$ | $71.03_{\pm1.14}$ | $94.68_{\pm0.22}$ | $91.03_{\pm0.24}$ | $93.42_{\pm0.17}$ | $89.00_{\pm0.39}$ |
| ✗ | ✗ | ✓ | $71.86_{\pm0.69}$ | $63.41_{\pm1.10}$ | $72.35_{\pm1.45}$ | $71.01_{\pm1.57}$ | $94.19_{\pm0.46}$ | $90.04_{\pm0.64}$ | $\mathbf{93.51}_{\pm0.29}$ | $\mathbf{89.21}_{\pm0.47}$ |
| ✓ | ✓ | ✗ | $73.83_{\pm0.71}$ | $65.54_{\pm1.16}$ | $72.68_{\pm1.42}$ | $71.43_{\pm1.42}$ | $95.11_{\pm0.29}$ | $91.63_{\pm0.45}$ | $93.45_{\pm0.28}$ | $89.02_{\pm0.46}$ |
| ✓ | ✗ | ✓ | $73.17_{\pm0.27}$ | $64.77_{\pm0.66}$ | $73.76_{\pm1.44}$ | $72.60_{\pm1.78}$ | $94.86_{\pm0.67}$ | $91.12_{\pm0.61}$ | $93.48_{\pm0.27}$ | $89.00_{\pm0.47}$ |
| ✗ | ✓ | ✓ | $73.90_{\pm0.84}$ | $66.10_{\pm1.10}$ | $73.98_{\pm1.62}$ | $72.93_{\pm1.68}$ | $93.95_{\pm0.16}$ | $89.81_{\pm0.38}$ | $74.41_{\pm16.4}$ | $64.24_{\pm21.4}$ |
| ✓ | ✓ | ✓ | $\mathbf{74.88}_{\pm0.49}$ | $\mathbf{67.18}_{\pm1.23}$ | $\mathbf{74.99}_{\pm0.31}$ | $\mathbf{73.82}_{\pm0.33}$ | $\mathbf{95.20}_{\pm0.18}$ | $\mathbf{91.66}_{\pm0.57}$ | $93.46_{\pm0.36}$ | $89.02_{\pm0.62}$ |

contributes to performance improvements, but the gains are relatively small when applied in isolation. In contrast, combining all three components consistently leads to the best result, demonstrating a clear synergistic effect. For Food-101, the individual components (W, WM, HS) do not yield additional performance gains, possibly because MIDAS leveraging misaligned samples already achieves strong performance. Overall, MIDAS clearly improves or is at least as good as the related baselines. These findings confirm that the components complement each other and that their integration is crucial to maximizing the effectiveness of learning from misaligned samples.

## 4.4 Comparison with Existing Data Augmentation Methods

In addition to imbalance-specific techniques, we also compare our method with modality-agnostic multimodal data augmentation strategies, including Mixup [37], PowMix [24], and LeMDA [27] on the CREMA-D and Food-101 datasets. While Mixup is originally designed for unimodal data, we extend it to the multimodal setting by interpolating each modality's embedding independently

Table 3: Performance evaluation of MIDAS against existing multimodal data augmentation strategies.

| Method | CREMA-D | | Food-101 | |
|---|---|---|---|---|
| | Acc (↑) | F1 (↑) | Acc (↑) | F1 (↑) |
| Joint training | $60.28_{\pm1.16}$ | $58.60_{\pm1.14}$ | $91.35_{\pm0.17}$ | $85.75_{\pm0.35}$ |
| Mixup [37] | $61.84_{\pm3.39}$ | $60.38_{\pm3.59}$ | $91.36_{\pm0.28}$ | $85.80_{\pm0.63}$ |
| PowMix [24] | $63.66_{\pm1.25}$ | $62.32_{\pm1.21}$ | $89.59_{\pm2.73}$ | $83.21_{\pm4.07}$ |
| LeMDA [27] | $58.13_{\pm2.74}$ | $56.75_{\pm2.45}$ | $91.19_{\pm0.21}$ | $85.56_{\pm0.43}$ |
| **MIDAS** | $\mathbf{74.99}_{\pm0.31}$ | $\mathbf{73.82}_{\pm0.33}$ | $\mathbf{93.11}_{\pm0.35}$ | $\mathbf{88.48}_{\pm0.60}$ |

and computing the label as a weighted average of the original labels, following the standard Mixup formulation. As shown in Table 3, MIDAS consistently outperforms data augmentation baselines in both accuracy and F1-score. These results indicate that typical multimodal data augmentation strategies are insufficient for handling modality imbalance, and they sometimes even show worse accuracy than Joint training. In contrast, MIDAS leverages misaligned samples with targeted weighting schemes, allowing the model to learn better from underutilized modalities. This demonstrates the importance of incorporating imbalance-aware design into multimodal data augmentation.

## 4.5 Additional Analysis

**Efficacy of weak-modality weighting** To validate the efficacy of the weak-modality weighting, we present the averaged normalized confidence scores of training pairs as training progresses on

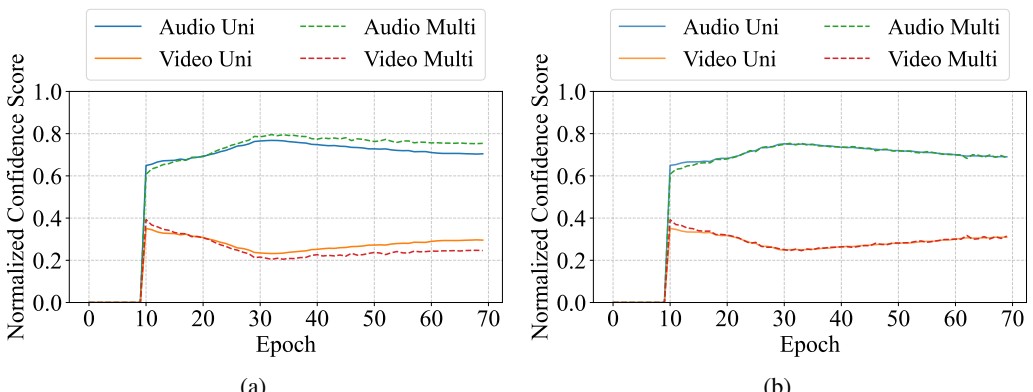

(a)                                                                              (b)

Figure 3: Normalized confidence score comparison between (a) MIDAS without the weak-modality weighting and (b) MIDAS with the weak-modality weighting on the CREMA-D dataset.

Table 4: Performance comparison between cosine similarity and L2 distance across three datasets.

| Method | Kinetics-Sounds | | CREMA-D | | Food-101 | |
|---|---|---|---|---|---|---|
| | Acc (↑) | F1 (↑) | Acc (↑) | F1 (↑) | Acc (↑) | F1 (↑) |
| L2 distance | 74.69 | 67.00 | 74.26 | 73.36 | 93.79 | 89.54 |
| Cosine similarity (Ours) | **75.26** | **68.34** | **75.00** | **74.05** | **93.82** | **89.56** |

the CREMA-D dataset. In Figure 3, "Uni" represents the confidence scores from the unimodal classifiers ($\mathbb{E}_{\tilde{x}_i \sim \tilde{B}}[\tilde{c}_i^m]$) and "Multi" represents the confidence scores from the multimodal classifier ($\mathbb{E}_{\tilde{x}_i \sim \tilde{B}}[(\tilde{c}_i)_{\tilde{y}_i^m}]$). The normalized confidence scores are measured when classifiers predict the misaligned training samples. As shown in Figure 3a, without the weak-modality weighting, the normalized confidence score gap of the multimodal classifier between audio (dominant modality) and video becomes larger than the target normalized confidence scores. The reason of the larger gap is that confidence-based labeling still assigns the larger value to the loss regarding the dominant modality, as we point out in Sec. 3.4. In contrast, with the weak-modality weighting, the normalized confidence scores of the unimodal classifiers and the multimodal classifier become more consistent within each modality, as shown in Figure 3b. This means that the weak-modality weighting leads the multimodal classifier to predict the misaligned samples closer to the target labels.

**Trend of weak-modality weight**   We analyze the trajectory of the weak-modality weight $\alpha$ throughout the training process on all datasets. As illustrated in Figure 4, after the warm-up phase, $\alpha$ increases to allocate greater weight to the undervalued modality. Notably, $\alpha$ converges as training progresses, indicating that it effectively captures the relative importance of each modality while maintaining an appropriate balance between them, without diverging.

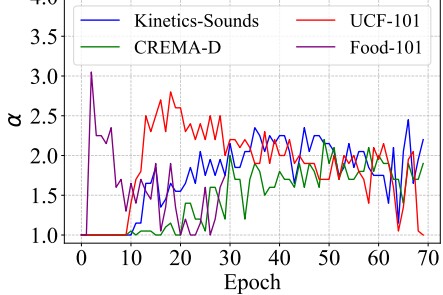

Figure 4: Trends of weak-modality weight $\alpha$ during training on four datasets.

**Similarity metric study for hard-sample weighting**   We compare cosine similarity for hard-sample weighting with L2 distance, a widely used distance metric. Cosine similarity is generally preferable for high-dimensional features due to its scale invariance and stronger discriminative properties [38, 39]. To empirically validate this choice, we conduct experiments where L2 distance replaces cosine similarity in our hard-sample weighting. As shown in Table 4, cosine similarity consistently outperforms L2 distance in terms of accuracy and F1-score. These results confirm that cosine similarity is more suitable for our hard-sample weighting based on high-dimensional feature similarity.

**Modality confidences during training**   We also provide modality confidence of Joint training and MIDAS when predicting misaligned validation data as training progresses on the CREMA-D dataset

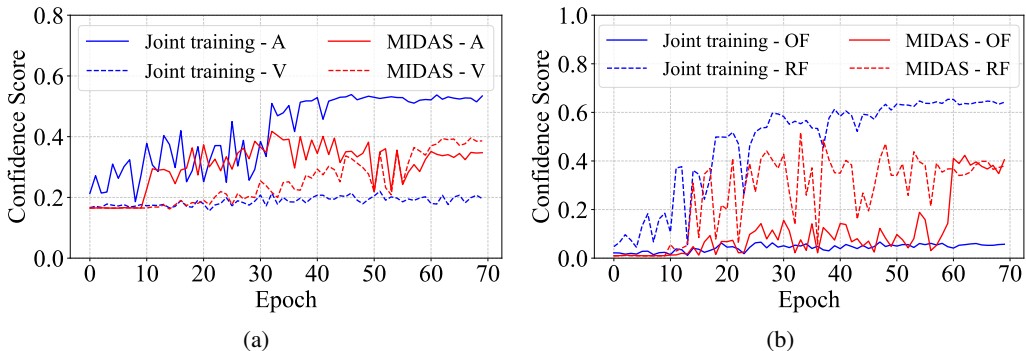

(a)                                    (b)

Figure 5: Model confidence curves of Joint training and MIDAS for each modality on (a) CREMA-D (audio, A; video, V) and (b) UCF-101 (optical flow, OF; RGB frame, RF) datasets.

and UCF-101 dataset. As shown in Figure 5, MIDAS predicts the misaligned samples based on two modalities in a balanced manner while Joint training solely relies on the audio. The additional confidence curves for the Kinetics-Sounds and Food-101 datasets are in the Appendix (Sec. A.4).

**Accuracy for misaligned samples** We show the accuracy of MIDAS on misaligned validation data across all datasets, compared to Joint training, in Figure 6. For all datasets, our method yields substantial improvements over Joint training, demonstrating its robustness under semantic inconsistency. This suggests that our method effectively captures modality-invariant representations, enabling the model to maintain reliable predictions even when semantic cues are partially misaligned.

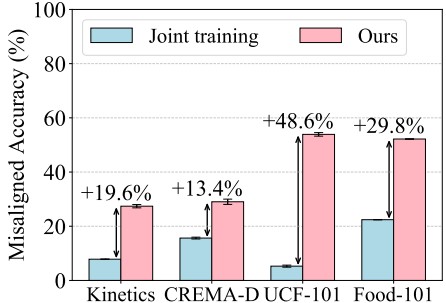

**Experiments on trimodal dataset** To demonstrate the scalability of our method, we further conduct experiments on the CMU-MOSI dataset [40], which contains three modalities: text, audio, and video. This dataset is widely used for multimodal sentiment analysis, consist-

Figure 6: Accuracy for misaligned validation samples comparison between Joint training and MIDAS.

ing of two classes: positive and negative. We use 1,284, 229, and 686 pairs for training, validation, and testing, respectively. We follow the experimental settings introduced by [6] and [41]. MIDAS achieves 74.00% accuracy and 73.64 F1-score, outperforming Joint training (71.13% accuracy and 70.86 F1-score). These results indicate that our method is not limited to simpler bimodal tasks but can effectively scale to more complex multimodal environments.

## 5 Conclusion

In this paper, we first identify misaligned samples as a key signal for exposing and addressing modality imbalance in multimodal learning. Based on this insight, we propose MIDAS, a novel data augmentation approach that constructs misaligned samples by pairing modalities from semantically different data points and labels them based on the confidence of unimodal classifiers. To strengthen the model's ability to extract useful information from these difficult samples, MIDAS incorporates two weighting mechanisms. First, weak-modality weighting amplifies the contribution of the least confident modality, thereby encouraging the model to utilize underrepresented signals during training. Second, hard-sample weighting prioritizes semantically ambiguous samples, making the model learn more effectively from misaligned samples. Extensive experimental results on various multimodal datasets demonstrate that MIDAS outperforms the baselines using both weighting strategies, confirming its effectiveness as a data-driven solution for mitigating the multimodal imbalance problem.

**Limitations:** While MIDAS demonstrates significant improvements in imbalanced multimodal learning, the current framework focuses primarily on classification tasks only. Extending applicability to other multimodal tasks, such as generation or retrieval, could require further adaptations.

## Acknowledgement

This material is based on work that is partially funded by an unrestricted gift from Google. This work was supported by the Institute of Information & Communications Technology Planning & Evaluation (IITP) grant funded by the Korea government (MSIT) (No. RS-2022-II220157, Robust, Fair, Extensible Data-Centric Continual Learning). This work was supported by the Institute of Information & Communications Technology Planning & Evaluation (IITP) grant funded by the Korea government (MSIT) (No. RS-2024-00444862, Non-invasive near-infrared based AI technology for the diagnosis and treatment of brain diseases).

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

# A  Appendix

## A.1  The Algorithm of MIDAS

---

**Algorithm 1:** The algorithm of MIDAS

---

**Input:** Training dataset $D$; Number of total epochs $E$; Warm-up epochs $E_w$, Batch size $b$;
       Learning rate $\gamma$; Hyperparameters $\lambda, \eta$; Model parameters
       $\theta = \{\varphi_1, \ldots, \varphi_M, g_f, g_1, \ldots, g_M\}$.

1   Initialize weak-modality weights $\alpha_m \leftarrow 1$ for $m = 1, \ldots, M$;
2   **for** *epoch = 1 to E* **do**
     `//Warm-up phase for training unimodal classifiers`
3     **if** *epoch < $E_w$* **then**
4        **for** *each mini-batch $B_{aligned} = \{(x_i, y_i)\}_{i=1}^b \subset D$* **do**
5           $p_i^m = \text{softmax}\,(g_m(\varphi_m(x_i^m)))$;
6           $\mathcal{L}_{\text{uni}} = \sum_{m=1}^M \mathcal{L}_{\text{CE}}(p_i^m, \mathbf{y}_i)$;
7           Update parameters $\theta \leftarrow \theta - \gamma \nabla_\theta \mathcal{L}_{\text{uni}}$;
8     **else**
        `//Main training`
9        Initialize $t = 1, \alpha_m^{(t)} = 1$ for $m = 1, \ldots, M$ ;
10       **for** $B = \{(x_i, y_i)\}_{i=1}^b \subset D$ **do**
        `//1.  Compute losses for aligned samples`
11         Compute $\mathcal{L}_{\text{uni}}(x_i, y_i)$;
12         Compute $\mathcal{L}_{\text{align}}(x_i, y_i)$ using Eq. 9;
        `//2.  Generate misaligned samples`
13         $\{x_j\}_{j=1}^b$ by random shuffling $B$ ;
14         $\tilde{B} = \{(\tilde{x}_i, \tilde{y}_i)\}_{i=1}^b$ using Eq. 2, 3;
        `//3.  Compute the loss for misaligned samples`
15         Compute $\mathcal{L}_{\text{mis}}(\tilde{x}_i, \tilde{y}_i; \alpha^{(t)}, \tilde{s}_i)$ using Eq. 10;
        `//4.  Compute the total loss and update parameters`
16         Compute $\mathcal{L}_{\text{total}}$ using Eq. 11;
17         Update parameters $\theta \leftarrow \theta - \gamma \nabla_\theta \mathcal{L}_{\text{total}}$;
        `//5.  Update weak-modality weight $\alpha$`
18         Identify batch-wise least confident modality $\hat{m}$ using Eq. 5;
19         Compute average update signal $\Delta_\alpha$ using Eq. 6;
20         Update $\alpha_{\hat{m}}^{(t+1)} \leftarrow \max(1, \alpha_{\hat{m}}^{(t)} + \eta \cdot \Delta_\alpha)$ using Eq. 7;
21         $t \leftarrow t + 1$;

**Output :** Trained model parameters $\theta$

---

## A.2  Generalization to M > 2 Modalities

Continuing from Sec. 3, while we focus on M=2 for clarity in the main paper, our approach readily generalizes to $M > 2$ modalities. The core mechanisms remain identical, with summations and operations extending naturally over $M$ modalities.

- **Generating misaligned samples** Generation involves selecting a set of $M$ source samples $K_i = \{i, j_1, \ldots, j_{M-1}\}$ with at least two different labels. A random permutation $\pi \in S_M$ determines the assignment $\tilde{x}_i = (x_{k_{(\pi(1))}}^1, \ldots, x_{k_{(\pi(M))}}^M)$, where $k_{(\cdot)}$ indexes into $K_i$.

- **Confidence based sample-level labeling** Confidences $(\tilde{p}_i^m)_{y_{k_{(\pi(m))}}}$ are computed for each modality $m$. Normalization uses $\sum_{l=1}^M (\tilde{p}_i^l)_{y_{k_{(\pi(l))}}}$ in the denominator. The soft label is $\tilde{y}_i = \sum_{m=1}^M \tilde{c}_i^m \mathbf{y}_{k_{(\pi(m))}}$.

Table 5: Summary of datasets used in our experiments.

| Dataset | #Train | #Val | #Test | #Class | Modality 1 | Modality 2 |
|---|---|---|---|---|---|---|
| Kinetics-Sounds | 16,890 | 2,461 | 4,778 | 31 | Audio | Video |
| CREMA-D | 5,209 | 744 | 1,489 | 6 | Audio | Video |
| UCF-101 | 9,159 | 1,308 | 2,618 | 101 | Optical Flow | RGB frame |
| Food-101 | 63,481 | 9,069 | 18,138 | 101 | Text | Image |

- **Weak-modality weighting** The least confident modality $\hat{m}$ is found using $\arg\min_{m\in\{1,\dots,M\}}$ and normalization involves $\sum_{l=1}^{M}$ in the denominator. The update signal $\Delta_\alpha$ and rule remain conceptually the same, targeting $\alpha_{\hat{m}}$.

- **Hard-sample weighting** The set of replaced modalities $R_i = \{m \mid k_{(\pi(m))} \neq i\}$ is identified. The weight $\tilde{s}_i$ is the average similarity $\frac{1}{|R_i|} \sum_{m\in R_i} \mathrm{sim}(f_i^m, f_j^m)$ where $y_j \neq y_i$ (if $R_i \neq \emptyset$).

## A.3 Experimental Details

**Dataset details**  Table 5 summarizes the number of samples in each split, class counts, and the types of modalities used for each dataset.

**Implementation details**  Continuing from Sec. 4.1, we provide additional configurations for implementing experiments. We use the SGD optimizer with momentum of 0.9 and an initial learning rate of 1e-3 for Kinetics-Sounds, CREMA-D, and Food-101, and 1e-2 for UCF-101. We use a batch size of 64 across all datasets. Models are trained for 30 epochs for Food-101 and 70 epochs for the other three datasets. We combine the features from different modalities by concatenation. The step size of the StepLR scheduler is 15 for Food-101 and 50 for the others. We apply a weight decay of 1e-4 and a StepLR learning rate schedule for all datasets. We use five workers for all experiments. For MIDAS, we use the hyperparameter $\lambda$ of 5 for the Kinetics-Sounds datasets, and 1 for others. We also provide an analysis of the hyperparameter $\lambda$ in Sec. A.4. We use $\eta$ of 5e-2 for all datasets. The best model is selected based on validation accuracy.

In practice, generating a misaligned sample at the input level can be computationally expensive as it requires passing each input component through its respective encoder for each modality. For efficiency, we implement this process at the feature level. Given a batch of samples $B = \{(x_i, y_i)\}$, we first compute the features $f_i^m = \varphi_m(x_i^m)$ for all $i$ and $m \in \{1, 2\}$. Then, we construct a misaligned feature vector $\tilde{f}$ by combining features from different samples within the batch:

$$\tilde{f}_i = (f_i^1, f_j^2) \tag{12}$$

where $y_i$ and $y_j$ are not identical. $\tilde{f}_i = (f_j^1, f_i^2)$ is also possible. This feature-level combination reuses the already computed features, significantly reducing the computational overhead compared to processing misaligned samples from scratch. Quantitative comparisons of the efficiency of feature-level augmentation are provided in Sec. A.4.

For the CMU-MOSI dataset, a trimodal dataset, we use Transformer encoders for all three modalities, trained from scratch. We employ the SGD optimizer with a momentum of 0.9 and a weight decay of 1e-4. The batch size is set to 64, and the models are trained for 70 epochs. A StepLR scheduler with a decay rate of 0.1 and a step size of 50 is used. The initial learning rate is 1e-2.

## A.4 Additional Experiments

**Modality confidence during training**  Continuing from Sec. 4.5, we additionally present the modality confidence curve on the Kinetics-Sounds and Food-101 datasets in Figure 7. Similar to Figure 5a, MIDAS utilizes all modalities in a much more balanced way when predicting multimodal data compared to the Joint training.

**Hyperparameter analysis**  As introduced in Sec. A.3, to further study the impact of the loss weight hyperparameter $\lambda$, which controls the contribution of the midaligned sample loss term $\mathcal{L}_{\mathrm{mis}}$, we

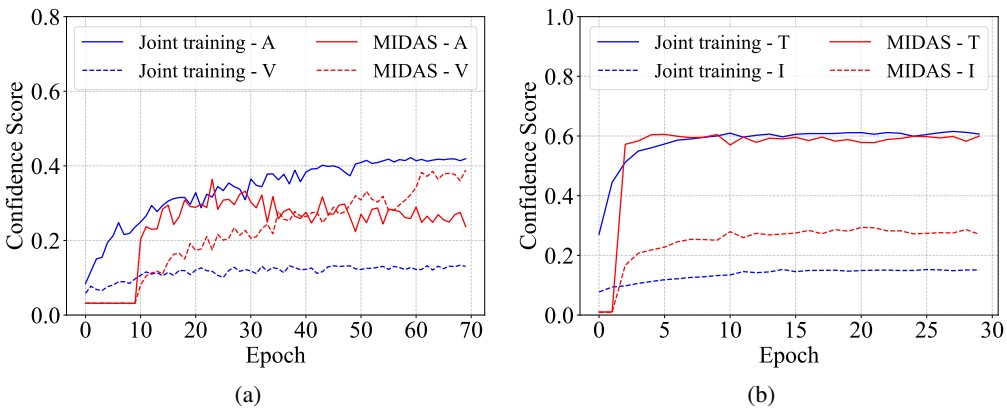

Figure 7: Model confidence curves of Joint training and MIDAS for each modality on the (a) Kinetics-Sounds (audio, A; video, V) and (b) Food-101 (text, T; image, I) datasets.

Table 6: Performance evaluation with varying $\lambda$ values across three datasets.

| $\lambda$ | Kinetics-Sounds | | CREMA-D | | Food-101 | |
|---|---|---|---|---|---|---|
| | Acc (↑) | F1 (↑) | Acc (↑) | F1 (↑) | Acc (↑) | F1 (↑) |
| 0.5 | 70.43 | 62.06 | 73.52 | 72.26 | 93.65 | 89.32 |
| 1 | 71.35 | 62.87 | **75.00** | **74.05** | **93.82** | **89.56** |
| 2 | 73.23 | 65.05 | 74.60 | 73.31 | 93.57 | 89.21 |
| 5 | 75.26 | **68.34** | 73.25 | 72.32 | 65.20 | 52.11 |
| 10 | **75.34** | 68.10 | 63.17 | 62.58 | n/a | n/a |
| 20 | 74.19 | 66.12 | 57.46 | 57.12 | n/a | n/a |

conduct an additional analysis on the Kinetics-Sounds, CREMA-D, and Food-101 datasets. As summarized in Table 6, the optimal $\lambda$ value varies depending on the dataset. When $\lambda$ is too small, the model under-utilizes the information from misaligned samples. Conversely, if $\lambda$ is too large, the model overly prioritizes misaligned samples and fails to adequately learn shared multimodal representations from aligned data.

**Evaluation on additional datasets**    To further validate the effectiveness of MIDAS, we additionally evaluate our method on the Sarcasm [42] and Twitter2015 [43] datasets, beyond the four widely used datasets included in Sec. 4. Both datasets consist of image and text modalities. The Sarcasm dataset contains 17,316 samples for training, 2,463 samples for validation, and 4,936 samples for test, while the Twitter2015 dataset includes of 3,736 pairs for training, 534 pairs for validation, and 1,068 pairs for test. As provided in Table 7, the results are consistent with our original findings, where MIDAS consistently outperforms competing methods across these datasets, confirming the same performance trend. These findings demonstrate that MIDAS is effective not only on standard benchmarks but also on diverse domains such as sarcasm detection.

**Efficiency of feature-level augmentation**    Feature-level augmentation leverages precomputed features after processing aligned samples. In contrast, input-level augmentation requires processing twice as many raw inputs (aligned and misaligned samples), thereby significantly increasing computational cost. To quantify this, we measure the training times on the CREMA-D, Kinetics-Sounds, and Food-101 datasets. As shown in Table 8, the feature-level method achieves substantial reductions in training time compared to the input-level method. These results confirm the efficiency advantages of our approach.

**Study of the misaligned sample generation methods**    To examine the effectiveness of the random replacement method in terms of model generalizability, we compare our method against a confusion-based replacement strategy, in which each sample is paired with another whose label is among its top-2 predicted classes. This alternative strategy makes misaligned samples inherently challenging or informative for better training. On the CREMA-D dataset, the confusion-based replacement yields an accuracy of 67.67% and an F1 score of 66.79. In contrast, our random replacement method achieves

Table 7: Performance comparison on the Sarcasm and Twitter2015 datasets.

| Method | Sarcasm | | Twitter2015 | |
|---|---|---|---|---|
| | Acc (↑) | F1 (↑) | Acc (↑) | F1 (↑) |
| Joint training | 86.89 | 86.38 | 71.82 | 64.35 |
| AMCo [12] | 85.92 | 85.39 | 71.44 | 58.59 |
| MCR [23] | 85.33 | 84.68 | 72.19 | 65.17 |
| **MIDAS (ours)** | **87.16** | **86.58** | **73.60** | **65.89** |

Table 8: Training time comparison (in minutes) between feature-level and input-level methods.

| Dataset | Feature-level (Ours) | Input-level |
|---|---|---|
| CREMA-D | 52 | 77 |
| Kinetics-Sounds | 311 | 445 |
| Food-101 | 278 | 693 |

higher performance, with an accuracy of 75.00% and an F1 score of 74.05. These results indicate that while confusion-based replacement enforces harder negative pairs, it limits generalization and ultimately reduces model performance. In comparison, random replacement provides a better balance between efficiency and generalizability, leading to stronger overall results.

**Role of modality balance in performance improvement**   To further investigate whether the performance gains of MIDAS stem from improved modality balancing, we evaluate the contributions of each modality using Shapley value-based modality valuation [13] on the CREMA-D dataset, which consists of audio and video modalities. Specifically, we compare predictions from models trained with joint training against those from MIDAS, while also considering unimodal baselines. For modality-specific analysis, we replace the features of the non-target modality with zero vectors and then measure performance in the usual way. For example, in an audio-only evaluation, the video features are replaced with zero vectors before computing the results.

The unimodal audio classifier achieves 60.0% accuracy, and the unimodal video classifier achieves 53.7%. When using joint training, the audio-only accuracy drops to 42.1%, and the video-only accuracy further drops to 19.2%, indicating severe under-utilization of the video modality. In contrast, MIDAS achieves much more balanced accuracy contributions with 54.3% for audio and 49.9% for video, leading to an overall accuracy of 73.7%. These results suggest that the substantial performance gain (from 60.2% with joint training to 73.7% with MIDAS) largely stems from improved modality balancing rather than a general regularization effect. Regularization alone cannot explain the drastic improvement in video feature utilization, which increases from 19.2% to 49.9%, while maintaining strong audio performance. This result confirms that MIDAS explicitly enhances modality balance, which is the key factor behind the observed performance improvements.

