# OpenReview forum: "MIDAS: Misalignment-based Data Augmentation Strategy for Imbalanced Multimodal Learning"
_NeurIPS.cc/2025/Conference — NeurIPS 2025 poster_

### Official Review · Reviewer_rnN8 · 2025-06-16

**Clarity:** 4
**Significance:** 3
**Originality:** 4
**Rating:** 5
**Confidence:** 3

**Summary:**

The paper presents a novel approach to addressing modality imbalance in multimodal learning by leveraging misaligned samples (synthetically generated examples that contain conflicting signals from different modalities). They posit that such samples force the model to use information from the non-dominant modality. They also propose two re-weighting mechanisms to further modality balance. One mechanism, weak-modality weighting, adjusts the contribution of the non-dominant modality in the loss computation based on the multi-model’s confidence. The second, hard-sample weighting increased the weight of samples where the substituted modality in the misaligned samples is similar to the original. Such samples are inherently harder for the model to learn, and hence provide a stronger signal for learning.

The proposed method is tested across several different multimodal datasets and shown to outperform other proposed methods for addressing modality imbalance. Ablations are performed to confirm that each each the three proposed components contributes meaningfully to the method’s efficacy.

**Questions:**

1. Do specific modality combinations or tasks make the proposed method more or less effective? This would provide greater insight into the generalizability of the approach.
2. Does the proposed method still offer performance benefits when the modalities are naturally balanced in their informational content? At the moment, it is unclear whether model performance could be degraded in such a scenario by training on misaligned samples.
3. Did you perform any experiments to specifically verify that the models actually do utilize information in both modalities more evenly after being trained with the proposed method? This seems important in validating the claim that MIDAS specifically addresses mobility imbalance.

**Ethical Concerns:**

["NO or VERY MINOR ethics concerns only"]

**Final Justification:**

I find this paper original, presented well, with findings backed by good experiments that were well explained/clarified including in the rebuttal. I hence have kept my original recommended score.

**Limitations:**

Yes

**Paper Formatting Concerns:**

None noted

**Quality:**

3

**Strengths And Weaknesses:**

Strengths:

The paper is well written and easy to follow;  the mathematical details of the proposed method are clearly outlined.

The results are well presented, and plots are easy to follow. The proposed method shows moderate performance gains over baselines.

The presented work is on a highly relevant topic given the rise of multimodal learning across various settings. It addresses an important gap in existing work (i.e., the lack of data-level approaches for handling modality imbalance).

The use of misaligned samples as a training signal in this context is a novel and interesting approach

Weaknesses:

No analysis is presented on how well the proposed method performs on different modality combinations.  Fig. 2 is a bit overly complex, difficult to understand. The performance results are only presented for bimodal datasets

No analysis is presented to demonstrate that improved modality balance is the actual mechanism behind the observed performance gains.

---

> ### Author Rebuttal · Authors · 2025-07-30
>
> We appreciate your insightful comments and suggestions. We would like to address your points.
>
> > W1: Analysis for diverse modality combinations
>
> We clarify that the results in Table 1 contain not only image-text but also other bimodal pairs (audio-video and RGB frame-optical flow), showing that our method consistently improves performance across diverse modality combinations. Moreover, in the Supplemental Material (Section B.1), we present additional experiments on a tri-modal dataset (text, audio, and video), further demonstrating that our approach generalizes effectively to multi-modality settings.
>
> > W2: Link between modality balance and performance gains
>
> We note that modality imbalance has been widely recognized as a primary bottleneck in multimodal learning. Numerous studies have shown that when multimodal models over-rely on dominant modalities, overall performance suffers [1, 2, 3]. Methods that explicitly rebalance modality contributions consistently yield substantial accuracy gains. MIDAS follows the same principle, and our experimental results align with the well-established connection between modality balance and multimodal performance.
>
> [1] Wang et al., What Makes Training Multi‑Modal Classification Networks Hard?, CVPR 2020
> [2] Huang et al., Modality Competition: What Makes Joint Training of Multi-modal Network Fail in Deep Learning? (Provably), ICML 2022
> [3] Xu et al., BalanceBenchmark: A Survey for Multimodal Imbalance Learning, ArXiv 2025
>
> > Q1: Modality-specific effectiveness
>
> The level of imbalance between modalities plays a major role in how much MIDAS improves performance. In our experiments (see Table 1), the audio-video modalities (Kinetics-Sounds and CREMA-D datasets) have significant imbalance, and MIDAS is thus very effective. On the other hand, the UCF-101 dataset uses RGB frame and optical flow modalities, which have lesser imbalance, so MIDAS is less effective, although it is still comparable to baselines. The imbalance varies across modality combinations as each modality has distinct characteristics [1].  We will discuss and cover these aspects in our revision.
>
> [1] Liang et al., Foundations and trends in multimodal machine learning: Principles, challenges, and open questions, ArXiv 2022
>
> > Q2: Naturally balanced modality setting
>
> When modalities are naturally balanced in their informational content, MIDAS should maintain performance without degradation for two key reasons. First, when modalities are naturally balanced, the unimodal confidence scores would be similar, leading to balanced weighting without artificial bias toward either modality. Second, the soft labels assigned to the misaligned samples (e.g., (0.5, 0.5)) are effectively true labels that help with regularization instead of being noisy. We will elaborate on these points in our revision.
>
> > Q3: Experiment for verifying improved modality balance
>
> Yes, we provide direct experimental validation through confidence tracking on the Kinetics-Sounds and CREMA-D datasets (Figures 1(b) and 4(b)). These experiments show that MIDAS produces more balanced confidence distributions across modalities compared to joint training, especially for misaligned samples. Under the assumption that a modality’s confidence reflects its actual contribution to the final decision, this balance in confidence implies that MIDAS is effectively leveraging information from each modality.

---

> ### Comment · Reviewer_rnN8 · 2025-08-05
>
> I thank the authors for their thoughtful response. They have largely addressed my outstanding questions. However, the link between modality balance and performance gains still remains unclear. True, modality imbalance is a major challenge in multi-modal learning. However, this does not necessarily follow that the mechanism by which the proposed method improves performance is modality balancing. It could, for example, simply act as a form of regularization that reduces overfitting.

---

> ### Author Response · Authors · 2025-08-06
>
> Thank you for this important clarification. To address your remaining question, we additionally analyze modality contributions in Joint training versus MIDAS by evaluating predictions with single modalities using Shapley value-based modality valuation [1] on the CREMA-D dataset (audio + video) (e.g., to perform audio-only analysis, we use a zero vector for video features and then measure performance as usual). The results are as follows:
>
> Metric         | Joint training | MIDAS (ours)
> ---------------|----------------|---------------
> Overall Acc    | 60.2           | 73.7
> Audio-only Acc | 42.1           | 54.3
> Video-only Acc | 19.2           | 49.9
>
> *Unimodal baselines: Audio classifier (60.0%), Video classifier (53.7%)
>
> Crucially, while the standalone video classifier achieves 53.7% accuracy, Joint training only obtains 19.2% accuracy using video features alone within the multimodal framework, indicating severe underutilization despite the video modality's inherent capability. MIDAS achieves much more balanced accuracy contributions (54.3% vs 49.9%) and effectively utilizes both modalities.
> This evidence suggests the multimodal performance gain (60.2% → 73.7%) largely stems from improved modality balancing rather than general regularization, as regularization alone would not explain the dramatic improvement in video feature utilization (19.2% → 49.9% accuracy) while maintaining strong audio performance. Furthermore, as shown in prior works [2, 3], regularization methods like dropout often fail to help or even worsen modality imbalance in imbalanced multimodal learning. We will clarify this point in our revision.
>
> [1] Wei et al., Enhancing Multimodal Cooperation via Sample-level Modality Valuation, CVPR 2024
> [2] Wang et al., What Makes Training Multi‑Modal Classification Networks Hard?, CVPR 2020
> [3] Wu et al., Characterizing and Overcoming the Greedy Nature of Learning in Multi-modal Deep Neural Networks, ICML 2022

---

### Official Review · Reviewer_TS6B · 2025-06-25

**Clarity:** 3
**Significance:** 2
**Originality:** 3
**Rating:** 4
**Confidence:** 4

**Summary:**

The paper presents MIDAS, a data augmentation strategy to artificially generate multimodal misaligned samples, i.e., samples with semantically inconsistent information across different modalities, for example, a picture of a car paired with the caption of a different object. MIDAS uses unimodal classification confidence to combine modalities from different samples. The network is then trained to match the unimodal classification performance with the multimodal classifier, encouraging the model to learn equally from both modalities, despite the misalignment.

**Questions:**

- Sec. 4.4 presents an analysis on the performance of MIDAS on misaligned data but few details are provided on this evaluation. How are the misaligned data obtained in this case? Given the focus of the paper on misaligned data, I think that the evaluation on misaligned data would require more attention.
- Are the baselines in the Experiments section also trained on the misaligned samples? If not, how would these baselines compare to MIDAS when trained on the same misaligned data? By looking at the first lines in Table 1 and Table 2, it seems the largest improvement is brought by training on misaligned data alone. It would also be useful to compare the same baselines in Table 1 in this setting.
- As the model is trained with a cross entropy loss, is it optimal to use cosine similarity for hard-sample weighting? I think it would be more appropriate to use cosine similarity if the classifier was a cosine classifier, and L2 distance when using cross entropy.

**Ethical Concerns:**

["NO or VERY MINOR ethics concerns only"]

**Final Justification:**

The paper's take on misaligned samples as an opportunity for multimodal learning is both interesting and novel, with potential for future extensions in realistic scenarios. The method is sound and validated on many datasets. The rebuttal and follow-up discussion clarified my doubts, mostly related to the misaligned samples generation pipeline and the evaluation setting. I am now confident in increasing the paper's score.

**Limitations:**

yes

**Quality:**

3

**Strengths And Weaknesses:**

Strengths:
- The paper focuses on semantic misalignment across different modalities in multimodal models and frames this scenario as an “opportunity” to encourage balanced learning across modalities. This approach is very intuitive but effective and proposes an interesting take on what might otherwise be seen as a problem.
- The experiments show consistent improvements compared to previous works, without requiring any modification to the architecture or using external data. Experiments focus on audio/video and image/text dataset, showing the effectiveness of the approach on different combinations of the input modalities.


Weaknesses:
- With MIDAS, the misalignment is created artificially by combining samples with different labels. However, semantic misalignment across modalities is a more broad issue that may arise naturally in data, for example due to annotation errors or intrinsic differences in the semantic information that each modality can encode [1]. In this sense, the proposed approach appears limited as it assumes that samples are normally aligned. Could it be extended to a more realistic scenario in which we don’t know if and how much the individual modalities composing a sample are misaligned? In other words, could MIDAS be applied to already misaligned samples?
- The labeling strategy and the weak-modality weighting mechanism in MIDAS encourage the multimodal classifier to match the confidence of the individual unimodal classifiers. However, different modalities may have different levels of information and convergence rates, making some modalities inherently more “informative” and therefore their respective unimodal classifiers more confident. This also makes the confidence scores not comparable across modalities. Why is the multimodal model expected to match the performance of their unimodal counterparts? Is this optimal? How would this compare with other approaches that “hardcode” the balance between the two modalities (for example giving the two modalities the same weight)?


[1] Von Kügelgen, Julius, et al. "Self-supervised learning with data augmentations provably isolates content from style." NeurIPS 2021

---

> ### Author Rebuttal · Authors · 2025-07-30
>
> We appreciate your constructive comments and would like to address your concerns.
>
> > W1: MIDAS in a realistic scenario
>
> MIDAS can be reasonably extended to handle cases where the degree of modality misalignment is unknown because we rely on the assumption that each unimodal classifier’s confidence score provides an appropriate proxy for alignment quality. If a modality is associated with incorrect or semantically confusing information, MIDAS will assign it a smaller weight, based on its lower confidence score obtained from the unimodal classifier. However, natural misalignment imposes fundamental limitations on all approaches for imbalanced multimodal learning, and MIDAS cannot completely resolve these unknown misalignments either. For example, a sample might contain an image of a cat and accompanying text “a kitten”, yet the label is mistakenly set to “dog” due to an annotation error. In this case, neither the vision nor the language classifier can confidently predict “dog”, so MIDAS assigns both modalities very low weights. Nevertheless, by combining confidence-based labeling with misaligned sample generation, MIDAS provides a practical mechanism for reducing their impact. We will explicitly clarify and discuss this potential extension in our revision.
>
> > W2: Optimality of unimodal classifiers
>
> We acknowledge that different modalities indeed have varying information levels and convergence rates, making direct confidence matching potentially suboptimal. However, in imbalanced multimodal learning, it is standard practice to leverage each modality's information signals across modalities simultaneously [1, 2, 3]. Accordingly, we employ a confidence-based approach that reflects each modality's actual information contribution, rather than enforcing an artificial balance.
> In addition, we clarify that our warm-up phase is specifically designed to mitigate the convergence rate differences by pre-training each unimodal classifier individually prior to joint multimodal training. This helps ensure that confidence scores are more comparable across modalities by allowing each to reach a stable state.
> Regarding hard-coded weighting approaches, we experiment on CREMA-D comparing our confidence-based method against equal (5:5) weighting:
>
> Method | Confidence-based (Ours) | hard coding (5:5)
> -------|-------------------------|-------------------
> Acc    |        **75.00**            |      70.23
> F1     |        **74.05**            |      69.02
>
> The results demonstrate that artificially balancing modalities regardless of their actual information content significantly harms performance. We will include these additional experimental comparisons in our revision to further validate this design choice.
>
> [1] Wei et al., Mmpareto: Boosting multimodal learning with innocent unimodal assistance, ICML 2024
> [2] Qian et al., Dyncim: Dynamic curriculum for imbalanced multimodal learning, ArXiv 2025
> [3] Peng et al., Balanced Multimodal Learning via On‑the‑Fly Gradient Modulation, CVPR 2021
>
> > Q1: Misaligned accuracy measurement
>
> To measure accuracy for misaligned samples, we generate misaligned data by randomly pairing modalities from different samples within the validation set, following the same procedure as for the training data, ensuring they contain conflicting semantic information. This synthetic misalignment strategy allows us to evaluate which modality the model relies on when faced with semantic inconsistencies. We consider the prediction correct if both labels from the sources of the misaligned samples are among the top-2 classes of the model's output.
> We acknowledge that this evaluation deserves a more detailed explanation, given its importance to our main contribution. In our revision, we will provide more comprehensive details about the data generation method, including specific examples of the misaligned pairs (e.g., pairing a "cat" image with "dog" text, where the prediction is considered correct if the model's top-2 output classes are "cat" and "dog").
>
> > Q2: Experimental setting details
>
> We clarify that the baselines are trained exclusively on aligned samples, as they do not consider misaligned samples for training. In contrast, MIDAS generates misaligned samples and leverages both aligned and misaligned samples during training, which is our main contribution. This represents a fair comparison since we evaluate each method under its optimal training conditions - baselines with aligned data, and MIDAS with its ability to benefit from additional misaligned samples. We will clarify this experimental setup more explicitly in our revision to avoid confusion.
>
> > Q3: Similarity metric for hard-sample weighting
>
> Compared to L2 distance, cosine similarity is generally preferable for high-dimensional features due to its scale invariance and better discriminative properties [1, 2, 3]. Our hard-sample weighting is also based on high-dimensional feature similarity, so we use cosine similarity. To further validate this choice, we perform new experiments below where we replace cosine similarity with L2 distance when using our method. As a result, using cosine similarity leads to better accuracy as expected. We will incorporate these experimental results in our revision to substantiate our choice.
>
> Metric | Dataset         | Cosine similarity (Ours) | L2 distance
> --------|------------------|---------------------------|-------------
> Acc     | Kinetics-Sounds  | **75.26**                     | 74.69
> F1      | Kinetics-Sounds  | **68.34**                     | 67.00
> Acc     | CREMA-D          | **75.00**                     | 74.26
> F1      | CREMA-D          | **74.05**                     | 73.36
> Acc     | Food-101         | **93.82**                     | 93.79
> F1      | Food-101         | **89.56**                     | 89.54
>
> [1] Radford et al., Learning Transferable Visual Models From Natural Language Supervision, ICML 2021
> [2] Robinson et al., Contrastive Learning with Hard Negative Samples, ICLR 2021
> [3] Kalantidis et al., Hard Negative Mixing for Contrastive Learning, NeurIPS 2020

---

> > ### Comment · Reviewer_TS6B · 2025-08-01
> >
> > I read the authors' rebuttal and the comments from other reviewers. I am mostly satisfied with the authors' responses and would like to ask a few clarifying questions.
> >
> > - Related to W2, I acknowledge that the warm-up phase mitigates the different convergence rates problem. However, modalities may still have different score distributions due to one modality being inherently more confident than the other. As also discussed in response to Q3 of rnN8, confidence scores converge to similar values across modalities when using MIDAS. Would it be theoretically possible to re-generate the misaligned samples during training using the updated models? This would ideally generate more difficult examples as the training progresses.
> >
> > - Related to Q1: are experiments in Fig. 4a evaluated on multiple runs? How stable are these numbers?
> >
> > - Related to Q2, I agree with the authors about the fairness of the comparison. As a follow up question, is the first row in table 2 the same model as 'Joint training' from Table 1 but with generated misaligned data? I would like to understand whether the introduction of misaligned samples alone is so effective.

---

> ### Author Response · Authors · 2025-08-02
>
> We appreciate your careful reading and would like to address your questions.
>
> > Q1: Combination of different modality confidence scores & Misaligned sample re-generation
>
> We clarify that our misaligned sample generation actually follows the reviewer's suggestion, where they are also re-generated during training. As detailed in our algorithm in the Appendix (Section A.1), after the warm-up phase, we regenerate misaligned samples on every mini-batch. This per-batch regeneration allows MIDAS to adapt and produce progressively harder samples. We realize we should have been more specific above.
> We agree that modalities may inherently differ in their confidence levels, and using warm-up is widely used to address this issue, not only in MIDAS, but also in many other approaches as a means to stabilize and calibrate modality-specific signals [1, 2]. Warm-up reduces the imbalanced score distributions and ensures that there is at least basic discriminative ability for each modality before modality valuation and reweighting [1].
>
> [1] Wei et al., Enhancing Multimodal Cooperation via Sample-level Modality Valuation, CVPR 2024
> [2] Huang et al., Adaptive Unimodal Regulation for Balanced Multimodal Information Acquisition, CVPR 2025
>
> > Q2: Stability of experimental results in Fig. 4a
>
> Yes, the results in Fig. 4a are evaluated over three random seeds. We include error bars representing the standard deviation for each result. The standard deviations are very small, so the error bars are barely visible. This indicates that the results are highly stable across multiple runs. For further clarity, we also provide the detailed averaged misaligned accuracies with standard deviations in the table below (same results as those in Fig. 4a). These results demonstrate that our method consistently achieves stable performance.
>
> Dataset         | Method         | Misaligned acc (mean) | Standard deviation
> ----------------|----------------|------------|---------------------
> Kinetics-Sounds | Joint training | 7.90       | 0.17
> Kinetics-Sounds | MIDAS          | 27.40      | 0.61
> CREMA-D         | Joint training | 15.64      | 0.34
> CREMA-D         | MIDAS          | 29.03      | 0.98
> UCF-101         | Joint training | 5.28       | 0.38
> UCF-101         | MIDAS          | 53.90      | 0.66
> Food-101        | Joint training | 18.60      | 5.37
> Food-101        | MIDAS          | 52.19      | 0.10
>
> > Q3: Effectiveness of using misaligned samples only
>
> Yes, the result in the first row of Table 2 demonstrates the effect of introducing misaligned samples without the warm-up phase or the two weighting mechanisms. This result was obtained using the same model as ‘Joint training’, but with additional unimodal classifiers used to label the misaligned samples, and the model was trained on both aligned and misaligned samples (i.e., model: same as Joint training + unimodal classifiers; training data: aligned + misaligned samples).
>
> As the results show, incorporating misaligned samples alone already yields a significant improvement over Joint training. However, this setup does not always outperform (or at least be on par with) more advanced baselines like SMV and MCR, which highlights the necessity of the proposed warm-up phase and the two weighting mechanisms.

---

> > ### Comment · Reviewer_TS6B · 2025-08-02
> >
> > I would like to thank the authors for clarifying my doubts. I had missed the error bars in Fig. 4a.

---

> > > ### Author Response · Authors · 2025-08-02
> > >
> > > Thanks, let us know if you have more questions.

---

### Official Review · Reviewer_Ny82 · 2025-06-29

**Clarity:** 3
**Significance:** 3
**Originality:** 4
**Rating:** 4
**Confidence:** 4

**Summary:**

This paper proposes MIDAS, a novel data augmentation strategy for imbalanced multimodal learning that leverages semantically misaligned samples instead of aligned ones. Specifically, MIDAS incorporates two weighting mechanisms to address existing challenges: (1) weak-modality weighting, which aims to better exploit underrepresented modalities, and (2) hard-sample weighting, which focuses on handling semantically ambiguous samples. Extensive experiments on multimodal classification tasks demonstrate the effectiveness of the proposed method.

**Questions:**

While the experimental evaluation is generally solid, this paper lacks comparisons with several advanced and competitive baselines, which limits the ability to fully assess the advantages of the proposed method. Moreover, the paper would benefit from visualizations illustrating how MIDAS’s two weighting mechanisms assign weights across modalities and samples, providing deeper insights into the model’s internal behavior. Additionally, it remains unclear whether the authors have explored the impact of alternative strategies—beyond random replacement—for generating misaligned samples on the method’s performance.

**Ethical Concerns:**

["NO or VERY MINOR ethics concerns only", "Major Concern: Improper research involving human subjects"]

**Final Justification:**

Thanks for the detailed rebuttal. I will keep my positive rating.

**Limitations:**

Yes.

**Paper Formatting Concerns:**

No.

**Quality:**

3

**Strengths And Weaknesses:**

Strengths:

(1)	This paper proposes MIDAS, a novel approach that leverages semantically misaligned samples to achieve superior performance in imbalanced multimodal learning.

(2)	The paper is well-written with a clear and logically structured presentation. The results are supported by well-designed figures and tables, which enhance the overall clarity of the work.

(3)	The experimental evaluation is relatively comprehensive, including comparisons with baselines, detailed ablation studies, and additional insightful discussions.

Weaknesses:

(1)	While the experimental evaluation is generally solid, it lacks comparisons with several advanced and competitive baselines (e.g., CGGM [1], Dyncim [2]), which limits a full assessment of the proposed method's advantages.

(2)	The paper would benefit from visualizations illustrating how MIDAS's two weighting mechanisms assign weights across modalities and samples, which could offer deeper insights into the model’s internal behavior.

[1] Guo Z, Jin T, Chen J, et al. Classifier-guided gradient modulation for enhanced multimodal learning[J]. Advances in Neural Information Processing Systems, 2024, 37: 133328-133344.

[2] Qian C, Han K, Wang J, et al. Dyncim: Dynamic curriculum for imbalanced multimodal learning[J]. arXiv preprint arXiv:2503.06456, 2025.

---

> ### Author Rebuttal · Authors · 2025-07-30
>
> We appreciate your thoughtful comments and would like to address your concerns.
>
> > W1/Q: Comparison with recent baselines
>
> We agree that a comparison with advanced baselines would further support the effectiveness of MIDAS. Therefore, we compare MIDAS to the advanced baselines (CGGM [1] and DynCIM [2]) on the CREMA-D dataset with three random seeds, as shown in the table below. As a result, MIDAS outperforms CGGM and DynCIM on average, even though both methods show strong results compared to the other baselines in our paper. For a specific seed, DynCIM shows better performance compared to MIDAS, but exhibits worse performance stability. We believe this is because both baselines rely on complex balancing mechanisms that combine signals across modalities. These mechanisms require careful tuning of gradient magnitudes or loss scaling factors during optimization. In contrast, MIDAS operates through sample-level labeling and weighting in the data space, providing more stable training dynamics without requiring sensitive gradient computations. We will discuss these points in our revision.
>
> Metric | CGGM             | DynCIM             | MIDAS (ours)
> --------|------------------|--------------------|-------------------
> Acc     | 68.99 ± 3.10     | 67.06 ± 8.51       | **74.99 ± 0.31**
> F1      | 67.21 ± 3.15     | 60.95 ± 11.05      | **73.82 ± 0.33**
>
> [1] Guo et al., Classifier-guided Gradient Modulation for Enhanced Multimodal Learning, NeurIPS 2024
> [2] Qian et al., DynCIM: Dynamic Curriculum for Imbalanced Multimodal Learning, ArXiv 2025
>
> > W2/Q: Visualization of two weighting mechanisms
>
> Thank you for this valuable suggestion. For weak-modality weighting, Fig. 3 illustrates how MIDAS dynamically adjusts the weak-modality weights during training across four datasets. The weights increase to allocate greater weight to the undervalued modality and converge as training progresses. For hard-sample weighting, we generate new histograms showing the distribution of similarity scores. For example, on the CREMA-D dataset, the histograms reveal that video samples exhibit a highly concentrated distribution (mean=0.875, std=0.093) with most samples being easy positives, while audio samples show a more dispersed distribution (mean=0.748, std=0.126) with greater variance, clearly demonstrating how MIDAS assigns higher weights to video samples with higher similarity scores. We will add the additional visualized analysis in our revision.
>
> > Q: Additional study of the misaligned sample generation method
>
> Thank you for this insightful idea. We use random replacement because of two key advantages: its computational efficiency and model generalizability. To illustrate how the model generalizes better, we compare random replacement against a “confusion-based” replacement strategy (pairing each sample with another whose label is among its top-2 predicted classes). This alternative strategy makes misaligned samples inherently challenging or informative for better training. As summarized in the table below, the results on the CREMA-D dataset indicate that random replacement outperforms confusion-based replacement. That is, training a model exclusively on hard samples limits its generalizability, leading to lower accuracy. We will include this comparative analysis and a brief discussion on alternative strategies in our revision.
>
> Metric | Confusion-based replacement | Random (Ours)
> --------|------------------------------|------------------
> Acc     | 67.67                        | **75.00**
> F1      | 66.79                        | **74.05**

---

> > ### Comment · Reviewer_Ny82 · 2025-08-06
> >
> > Thanks for the detailed rebuttal. I will keep my positive rating.

---

### Official Review · Reviewer_W7Kn · 2025-07-01

**Clarity:** 3
**Significance:** 3
**Originality:** 3
**Rating:** 4
**Confidence:** 4

**Summary:**

This paper introduces MIDAS (Misalignment-based Data Augmentation Strategy), an approach addressing modality imbalance in multimodal learning. The method generates semantically inconsistent cross-modal samples (e.g., pairing a "cat" image with "dog" text) labeled using unimodal confidence scores, compelling models to learn from conflicting signals. MIDAS incorporates two weighting mechanisms: weak-modality weighting enhances learning of underrepresented modalities through dynamic loss adjustment, while hard-sample weighting prioritizes semantically ambiguous samples to improve discriminative power.

**Questions:**

Please see weaknesses.

**Ethical Concerns:**

["NO or VERY MINOR ethics concerns only"]

**Final Justification:**

Most of my concerns are solved except for the theoretical foundation. But considering the methodology, I'd like to raise my score.

**Limitations:**

Yes

**Quality:**

3

**Strengths And Weaknesses:**

Strengths
1. The use of misaligned samples as supervised training signals, along with dynamic weighting mechanisms, introduces a new perspective.
2. The paper conducts thorough experiments on four multimodal datasets, with results showing improvements over existing methods.
3. The methodology is described systematically, with clear mathematical formulations and supporting diagrams.

Weaknesses
1. The impact of key hyperparameters (e.g., loss weight λ) is not systematically studied.
2. The experimental validation remains limited in scope, failing to include several key benchmark datasets such as Sarcasm, Twitter2015, NVGesture, and VGGSound.
3. Although empirically effective, the paper lacks a theoretical analysis for the proposed weighting schemes, which could deepen understanding of their behavior.
4. While feature-level augmentation is mentioned as an efficiency measure, no quantitative comparison (e.g., training time, memory usage) is provided to support this claim.

---

> ### Author Rebuttal · Authors · 2025-07-30
>
> We appreciate your helpful comments and would like to address your concerns.
>
> > W1: Hyperparameter study
>
> Following your great suggestion, we conduct an additional analysis to study the impact of the loss weight hyperparameter ($\lambda$) on the Kinetics-Sounds, CREMA-D, and Food-101 datasets. As summarized below, the optimal $\lambda$ value varies depending on the dataset. When $\lambda$ is too small, the model under-utilizes the information from misaligned samples. Conversely, if $\lambda$ is too large, the model overly prioritizes misaligned samples and fails to adequately learn shared multimodal representations from aligned data. We will discuss the hyperparameter study in our revision.
>
> Metric | Dataset         | 0.5   | 1     | 2     | 5     | 10    | 20
> --------|------------------|-------|-------|-------|-------|-------|-------
> Acc     | Kinetics-Sounds  | 70.43 | 71.35 | 73.23 | 75.26 | **75.34** | 74.19
> F1      | Kinetics-Sounds  | 62.06 | 62.87 | 65.05 | **68.34** | 68.10 | 66.12
> Acc     | CREMA-D          | 73.52 | **75.00** | 74.60 | 73.25 | 63.17 | 57.46
> F1      | CREMA-D          | 72.26 | **74.05** | 73.31 | 72.32 | 62.58 | 57.12
> Acc     | Food-101         | 93.65 | **93.82** | 93.57 | 65.20 |  n/a  |  n/a
> F1      | Food-101         | 89.32 | **89.56** | 89.21 | 52.11 |  n/a  |  n/a
>
>
> > W2: Additional benchmarks
>
> In addition to the four widely used benchmarks [1] already included in our paper, we appreciate the suggestion and will further evaluate MIDAS against baselines on the Sarcasm and Twitter2015 datasets. The results are consistent with our original findings, with MIDAS outperforming competing methods and confirming the same performance trend, demonstrating that MIDAS is effective on diverse benchmarks.  We will include these additional results in our revision.
>
> Metric | Dataset     | Joint training | AMCo   | MCR    | MIDAS (ours)
> --------|-------------|----------------|--------|--------|---------------
> Acc    | Sarcasm     | 86.89          | 85.92  | 85.33  | **87.16**
> F1     | Sarcasm     | 86.38          | 85.39  | 84.68  | **86.58**
> Acc    | Twitter2015 | 71.82          | 71.44  | 72.19  | **73.60**
> F1     | Twitter2015 | 64.35          | 58.59  | 65.17  | **65.89**
>
>
> [1] Xu et al., BalanceBenchmark: A Survey for Multimodal Imbalance Learning, ArXiv 2025
>
> > W3: Theoretical analysis
>
> We acknowledge that our weighting schemes primarily rely on empirical justification, but the weak-modality weighting strategy is theoretically motivated by the modality competition problem analyzed by [1]. This strategy also conceptually aligns with gradient-based [2], confidence-based [2], and loss-based [3] balancing methods, which predominantly depend on empirical validation. Our hard-sample weighting draws upon robust empirical evidence showing that challenging samples consistently improve model training across diverse domains [4, 5]. While the theoretical characterization of this phenomenon remains an active research area, our systematic empirical validation provides practical insights that advance the field's understanding. We appreciate the reviewer’s suggestion and will clarify these points in our revision.
>
> [1] Huang et al., Modality Competition: What Makes Joint Training of Multi-modal Network Fail in Deep Learning? (Provably), ICML 2022
> [2] Peng et al., Balanced Multimodal Learning via On‑the‑Fly Gradient Modulation, CVPR 2021
> [3] Wang et al., What Makes Training Multi‑Modal Classification Networks Hard?, CVPR 2020
> [4] Robinson et al., Contrastive Learning with Hard Negative Samples, ICLR 2021
> [5] Kalantidis et al., Hard Negative Mixing for Contrastive Learning, NeurIPS 2020
>
> > W4: Efficiency of feature-level augmentation
>
> We agree that quantitative comparisons would strengthen our claims about efficiency. Feature-level augmentation leverages precomputed features after processing aligned samples. In contrast, input-level augmentation requires processing twice as many raw inputs (aligned and misaligned samples), thereby significantly increasing the computational cost. To quantify this, we measure the training times on the CREMA-D, Kinetics-Sounds, and Food-101 datasets as shown in the table below. We appreciate your suggestion and will explicitly include detailed training-time comparisons in our revision.
>
> Dataset         | Feature-level (Ours) | Input-level
> ----------------|----------------------|-------------
> CREMA-D         | 52 min                   | 77 min
> Kinetics-Sounds | 311 min                  | 445 min
> Food-101        | 278 min                  | 693 min

---

> > ### Comment · Reviewer_W7Kn · 2025-08-07
> >
> > Thanks for the reply. Most of my concerns are solved except for the theoretical foundation. But considering the methodology, I'd like to raise my score.

---

### Note · Authors · 2025-08-13

We sincerely appreciate the reviewers and ACs for their thoughtful feedback and constructive discussions.
In this work, we propose MIDAS, a novel data augmentation strategy for imbalanced multimodal learning that semantically generates misaligned samples to boost underrepresented modalities using two weighting mechanisms. Extensive experiments show MIDAS consistently outperforms baselines across diverse modalities, demonstrating its effectiveness and generality.
Throughout our rebuttal and discussions with reviewers, we addressed concerns through comprehensive additional analyses and experiments. These included hyperparameter studies, additional benchmark results, advanced baseline comparisons, efficiency evaluations, and alternative similarity metrics, which confirmed MIDAS's effectiveness and resulted in maintained or improved positive ratings.
Thank you again for the opportunity to present and discuss our work. We hope our additional analyses and clarifications have been helpful in addressing your thoughtful feedback.

---

### Decision · Program_Chairs · 2025-09-17

**Decision:**

Accept (poster)

**Comment:**

The paper proposes to generate misaligned samples as a data augmentation strategy to handle modality imbalance problem in multimodal learning. Unimodal confidence scores are leveraged to achieve this generation and further a weighting mechanism is introduced to support weaker modalities during learning.

Some notable strengths of the submission include:
- a novel data augmentation strategy to generate misaligned samples
- thorough experimental results and the paper is well-written

Among weaknesses, the concerning were:
- missing impact on hyperparameters
- evaluation scope is limited, as some important benchmarks are missing (e.g., Sarcasm, Twitter2015, NVGesture)
- some visualizations are required to better understand the two weighting mechanisms
- grounding of the method that proposed modality balanced actually provides the claimed gains
- assumption of data being normally aligned and what if the data is by default misaligned

After the post-rebuttal discussion phase, majority of the reviewers remained positive about the novelty of the submission and the experimental validation. Some new results such as hyperparameter study, more dataset results, comparison with more recent baselines were provided during the rebuttal and post-rebuttal discussion period. Furthermore, the clarifications on normal aligned data assumption and stability of experimental results were acknowledged by the reviewers. Overall, nothing major or concerning remained after the closing of author-reviewer discussion. Therefore, the AC decides to recommend acceptance.